# GINA Implementation Improves Asthma Symptoms Control and Lung Function: A Five-Year Real-World Follow-Up Study

**DOI:** 10.3390/jpm13050809

**Published:** 2023-05-10

**Authors:** Nguyen Van Tho, Vu Tran Thien Quan, Do Van Dung, Nguyen Hoang Phu, Anh Tuan Dinh-Xuan, Le Thi Tuyet Lan

**Affiliations:** 1Department of Tuberculosis and Lung Diseases, Faculty of Medicine, University of Medicine and Pharmacy at Ho Chi Minh City, Ho Chi Minh City, Vietnam; thonguyen0225@ump.edu.vn; 2Department of Pulmonary Functional Exploration, University Medical Center, University of Medicine and Pharmacy at Ho Chi Minh City, Ho Chi Minh City, Vietnam; vutranthienquan@gmail.com; 3Department of Pathophysiology-Immunology, Faculty of Medicine, University of Medicine and Pharmacy at Ho Chi Minh City, Ho Chi Minh City, Vietnam; 4Department of Biostatistics, Faculty of Public Health, University of Medicine and Pharmacy at Ho Chi Minh City, Ho Chi Minh City, Vietnam; dovandzung@gmail.com; 5Department of Pulmonology, Dong Nai General Hospital, Bien Hoa City, Vietnam; nguyenhoangphu.vn@gmail.com; 6AP-HP, Hôpital Cochin, Service de Physiologie-Explorations Fonctionnelles, Paris, France; anh-tuan.dinh-xuan@aphp.fr

**Keywords:** asthma, GINA, lung function, pulmonary function tests, spirometry

## Abstract

Symptoms control remains challenging for most patients with asthma. This study was conducted to evaluate the level of asthma symptoms control and lung function over 5 years of GINA (Global INitiative for Asthma) implementation. We included all patients with asthma who had been managed following GINA recommendations at the Asthma and COPD Outpatient Care Unit (ACOCU) of the University Medical Center in Ho Chi Minh City, Vietnam from October 2006 to October 2016. Of 1388 patients with asthma managed following GINA recommendations, the proportion of patients with well-controlled asthma significantly improved from 2.6% at baseline to 66.8% at month 3, 64.8% at year 1, 59.6% at year 2, 58.6% at year 3, 57.7% at year 4, and 59.5% at year 5 (*p* < 0.0001 for all comparisons). The proportion of patients with persistent airflow limitation significantly decreased from 26.7% at baseline to 12.6% at year 1 (*p* < 0.0001), 14.4% at year 2 (*p* < 0.0001), 15.9% at year 3 (*p* = 0.0006), 12.7% at year 4 (*p* = 0.0047), and 12.2% at year 5 (*p* = 0.0011). In patients with asthma managed according to GINA recommendations, asthma symptoms control and lung function improved after 3 months and the improvement was sustained over 5 years.

## 1. Introduction

The Global Initiative for Asthma (GINA) has published recommendations to help physicians diagnose and manage patients with asthma properly [1]. These recommendations underline the importance of treating stable asthma by using controller medications to control symptoms and prevent future poor outcomes such as exacerbations, lung function decline, and medication side effects [1]. In practice, a substantial proportion of patients with asthma are poorly controlled because of suboptimal treatment [1]. A study on 2467 patients with asthma in 8 Asia-Pacific countries in 2014–2015 showed that only 17.8% were well-controlled according to GINA criteria [2]. In 2011, the Asthma Insight and Management survey conducted in 8 Asia-Pacific countries showed that asthma had a profound impact on patients’ health and quality of life: 64% reported exacerbations in the past 12 months; 66% reported missing work or school for asthma during the past year [3]. There is evidence that poorly controlled asthma is associated with urgent healthcare utilization [4], which gives rise to higher medication costs and a substantial reduction in quality of life. On the contrary, good asthma control saved the total cost of asthma management [5].

Since 2000, we have implemented the GINA strategy at the Asthma and COPD Outpatient Care Unit (ACOCU) of the University Medical Center in Ho Chi Minh City (UMC-HCM). We have transferred this model of ACOCU to other hospitals, especially district-level hospitals throughout Vietnam. A one-year follow-up study at 4 ACOCUs in Ho Chi Minh City showed that the proportion of patients with GINA-defined well-controlled asthma rose from 1.0% to 36.8%, and average FEV_1_ (forced expiratory volume in one second) increased from 71.8% to 79.7% of predicted values [6]. We consequently hypothesized that improvements in control of asthma symptoms and lung function are sustained over several years if patients consistently kept receiving appropriate asthma treatment. In order to verify this hypothesis we conducted the present study to evaluate the proportion of patients with controlled asthma symptoms and improved lung function over a five-year period of GINA implementation.

## 2. Patients and Methods

### 2.1. Study Design and Setting

This is a retrospective, observational, and real-world follow-up study. This study was conducted at the ACOCU of UMC-HCMC, a tertiary teaching hospital in southern Vietnam. This unit functions as a model for asthma management in the community in Vietnam [6]. All doctors and nurses from this unit had attended a two-weeks hands-on training course on the management of asthma and COPD in the community following GINA and GOLD (Global Initiative for Obstructive Lung Disease) recommendations. In this hands-on training course, participants were trained to perform spirometry, interpret spirometry results, diagnose and classify asthma, counsel patients with asthma, and prescribe asthma controllers and relievers following GINA recommendations [6]. Patients with asthma attending this unit had to pay by themselves or co-pay with health insurance for asthma medications depending on whether or not they had a referral letter from lower-level hospitals.

Patients with asthma had been managed by attending doctors following GINA recommendations. In short, patients with stable asthma had been prescribed asthma controllers based on a stepwise approach depending on their level of symptom control and spirometry results [7]. The asthma controllers consisted of an inhaled corticosteroid (ICS) (fluticasone propionate) or the combination of an ICS and a long-acting inhaled beta_2_-agnosit (LABA) (either fluticasone propionate/salmeterol or budesonide/formoterol) or montelukast [7]. At the first visit, all patients underwent a complete medical history interview and physical examination. Plain chest X-rays were performed to exclude infectious lung diseases. Patients performed spirometry using the KoKo^®^ spirometer (nSpire Health, Inc., Longmont, CO, USA) before and 15 min after inhalation of 400 microgram of salbutamol. The spirometry maneuvers met the American Thoracic Society/European Respiratory Society quality standard criteria [8]. Spirometry parameters were calculated as a percentage of predicted values based on the reference equation of NHANES III with an adjustment factor of 0.88 for Asians [9]. At each visit, patients were evaluated for level of asthma symptoms control, asthma severity according to the criteria of GINA 2006 [7], asthma controllers adherence and side effects. Patients performed pre-bronchodilator spirometry when needed at the follow-up visits.

### 2.2. Patients

Eligible subjects were all patients with asthma who were older than 12 years of age attending the ACOCU of UMC-HCM from October 2006 to October 2016. Patients were included if they fulfilled all the following criteria: a definitive diagnosis of asthma had been previously established, based on clinical symptoms such as recurrent attacks of cough, wheezing, chest tightness, and shortness of breath in the early morning or when exposed to the triggering factors; asthma symptoms had occurred more than one year before the enrolment; the first visit to the unit must have taken place from October 2006 to October 2011; and having had at least 2 visits to the unit. Patients were excluded from the study if they had any of the following criteria: pregnancy; breast-feeding; other chronic diseases that affected the quality of life such as active pulmonary tuberculosis, all types of cancer, congestive heart failure, kidney failure, and liver failure.

### 2.3. Data Collection

Data were extracted from patients’ medical records which were kept in two forms: the electronic form had been saved on the server of the hospital and the paper form had been kept in a storage room. All the patients were followed up for 5 years and were censored when they stopped to attend the unit or until October 2016. During the follow-up period, patients might have several visits which might not take place at the predetermined time; consequently, only the visit which was closest to the predetermined time was chosen. The times of follow-up visits were preset at 3, 6, 12, 24, 36, 48, and 60 months after enrolment for the seven visits (2 to 8, respectively) following baseline assessment.

Data of all patients were extracted from the electronic medical records using an in-house extraction program. Paper-based medical records of all patients who met the inclusion criteria were retrieved from the storage room. Data from the electronic medical records were extracted on a spreadsheet of an Excel file. Missing data from the electronic medical records were looked for in the paper-based medical records to fill out the variables on the Excel file.

Study variables included clinical and spirometry data as well as asthma therapies. Level of asthma symptoms control was classified according to the criteria of GINA 2006 (Appendix A) for visits happening before 2012 [7] and reclassified according to GINA 2016 whose criteria are the same as GINA 2022 (Appendix A) [1] for the purpose of comparisons with other published studies [10,11,12]. FEV_1_ and/or PEF (peak expiratory flow) are included in the GINA 2006 [7] but are not in the GINA 2016 criteria [1]. Persistent airflow limitation was defined as having both FEV_1_/FVC (forced vital capacity) < 0.70 and FEV_1_ < 80% of predicted values [13]. The persistent airflow limitation is a component of asthma evaluation because it is associated with adverse future outcomes [1] or chronic obstructive pulmonary disease with asthma features [14,15]. High-dose ICS was defined as fluticasone propionate > 500 mcg or budesonide > 800 mcg [1].

### 2.4. Statistical Analysis

Comparison of the proportion of patients with well-controlled asthma or with persistent airflow limitation or with FEV_1_ < 60% or on a high dose of ICS between each follow-up visit and the baseline visit was made using McNemar’s test. Comparisons of the number of annual asthma-related exacerbations or hospitalizations between each follow-up visit and the baseline visit were examined by using paired Wilcoxon signed rank test. Comparisons of spirometry parameters between each follow-up visit and the baseline visit were made by paired Student-*t*-test. Differences between groups were examined using a Chi-square test or Wilcoxon test, as appropriate. A *p*-value < 0.05 was considered statistically significant. Statistical analysis was done by using JMP 9.0.2 software (SAS Institute Inc., Cary, NC, USA). Some results of this study were previously published in abstract form [16].

## 3. Results

### 3.1. Baseline Characteristics

Among 4871 patients with asthma extracted from the electronic medical records, 1388 patients (28.5%) met all the inclusion criteria (Figure 1). The age of the included patients ranged from 12 to 89. Patients attending the ACOCU of UMC-HCM came from 39 provinces in Vietnam, 456 patients (32.9%) were from Ho Chi Minh City. Table 1 presents the characteristics of patients with asthma at the baseline visit.

### 3.2. Level of Asthma Symptoms Control

Over 5 years of asthma management, the proportion of patients with well-controlled asthma significantly improved from 2.6% at baseline to 66.8% at month 3, 69.6% at month 6, 64.8% at year 1, 59.6% at year 2, 58.6% at year 3, 57.7% at year 4, and 59.5% at year 5 (*p* < 0.0001 for all comparisons between each of the follow-up visits and the baseline visit) (Figure 2). The improvement of asthma symptoms control occurred as early as at three months and was sustained for 5 years. Although we were able to assess asthma symptoms control in only 121 patients who attended the follow-up visit at year 5, the results were comparable to those of 1329 patients with asthma at the baseline visit (Appendix A).

There was no relation between baseline characteristics and the proportion of patients with well-controlled asthma at year 1 or at year 5 (Appendix A, respectively).

The annual rate of hospitalizations for asthma decreased significantly at year 1, year 2, year 3, and year 4 compared with the previous year of the baseline visit (Table 2). Similarly, the annual rate of asthma exacerbations decreased significantly at year 2, year 4, and year 5 compared with the previous year of the baseline visit (Table 2).

### 3.3. Lung Function

Spirometry parameters including FVC, FEV_1_, FEV_1_/FVC, and PEF improved significantly and consistently at year 1, year 2, year 3, year 4, and year 5 from the baseline visit (Table 3).

The proportion of patients with persistent airflow limitation significantly decreased from 26.7% at baseline to 12.6% at year 1 (*p* < 0.0001), 14.4% at year 2 (*p* < 0.0001), 15.9% at year 3 (*p* = 0.0006), 12.7% at year 4 (*p* = 0.0047), and 12.2% at year 5 (*p* = 0.0011) (Figure 3). Similarly, the proportion of patients with FEV_1_ < 60% significantly decreased from 18.5% at baseline to 7.4% at year 1 (*p* < 0.0001), 7.3% at year 2 (*p* < 0.0001), 9.2% at year 3 (*p* = 0.0006), 7.3% at year 4 (*p* = 0.0073), and 6.5% at year 5 (*p* = 0.0455) (Figure 3). The difference in the proportion of patients with persistent airflow limitation or with FEV_1_ < 60% was not statistically significant between year 1 and the following years (*p* > 0.05).

### 3.4. ICS Dosage and Rate of Follow-Up Visits

The proportion of patients on high-dose ICS (equivalent fluticasone propionate > 500 µg/day) significantly decreased from 79.2% at baseline to 31.4% at year 1 (*p* < 0.0001), 31.7% at year 2 (*p* < 0.0001), 35.6% at year 3 (*p* < 0.0001), 39.5% at year 4 (*p* < 0.0001), and 27.2% at year 5 visit (*p* < 0.0001) (Appendix A).

Among 1388 patients included, the number of patients who attended follow-up visits at year 1, year 2, year 3, year 4, and year 5 was 777 (56.0%), 436 (31.4%), 309 (22.3%), 197 (14.2%), and 128 (9.2%), respectively (Figure 1). In the bivariate analysis, the following baseline characteristics were positively associated with the proportion of patients who attended the follow-up visit at year 1: residing in other provinces (*p* = 0.0178); having allergic rhinitis (*p* = 0.0059); BMI < 25 kg/m^2^ (*p* = 0.0219); having co-payment by health insurance (*p* = 0.0462); receiving fluticasone propionate (*p* = 0.0111) (Appendix A). In multivariate logistic regression analysis, only allergic rhinitis was independently associated with the proportion of patients who attended the follow-up visit in year 1 (Table 4).

## 4. Discussion

This real-world study showed that if patients with asthma were managed appropriately, the proportion of well-controlled asthma rose significantly, lung function increased significantly, and the proportion of persistent airflow limitation or of FEV_1_ < 60% fell significantly. These improvements occurred after 3 months and were sustained over 5 years of asthma management. These results corroborate the findings from our previous study which showed that the levels of asthma control and lung function improved after 3 months and were sustained over one year [6]. Another study showed that in asthma patients with regular ICS treatment, FEV_1_ started to improve within days, and reached a plateau after around 2 months [17]. The improvements in asthma symptoms control and lung function are associated with a decreased annual rate of hospitalizations for asthma or asthma exacerbations as shown in this study or in the GOAL study [12].

The finding that there was no difference in baseline characteristics between patients with well-controlled and patients with uncontrolled or partly controlled at year 1 or at year 5 suggests that about 60% of patients with asthma in the real world could be well-controlled if they were managed appropriately over time by attending physicians who followed GINA recommendations. This is in line with the results of a post-hoc analysis of the GOAL study which showed that the proportion of patients achieving well-controlled asthma according to GINA 2016 criteria ranged from 40–64% after one year of receiving ICS/LABA [12]. Well-controlled asthma is associated with significantly lower healthcare resource utilization, lower total cost, and higher quality of life [10]. The proportion of controlled asthma in this study was greater than in other previous studies which might be related to the criteria of well-controlled asthma applied or the asthma medications the patients received. In the original GOAL study and our previous study, the proportion of well-controlled was 41% [18] and 36.8% [6], respectively. In previous studies, the level of asthma control was defined by GINA 2006 criteria [7] which were more stringent (which includes FEV_1_ and/or PEF as one of the criteria) than GINA 2016 criteria [1]. The proportion of well-controlled asthma in this study was greater than that in surveys in the Asia Pacific region in 2011 (2%) [3] or in 2014 (17.8%) [2]. In those studies, patients with asthma were from the community and an average of 62% of patients used asthma controllers [3] or 13.9% used controllers regularly during the past week [2]. In this study, attending doctors followed GINA recommendations to manage patients with asthma more properly.

Most (96.8%) patients with asthma in this study received ICS/LABA as asthma controllers at the baseline visit because 93.2% of patients were moderate or severe persistent asthma. The reasons why no patient received stand-alone ICS at the baseline visit may be as follows: patients with asthma only visited the clinic when they were severe enough that attending doctors had to initiate asthma controllers at step 3; stand-alone ICS was not available at the clinic when attending doctors prescribed asthma controllers; and attending doctors might prefer ICS/LABA combination to stand-alone ICS. The finding that fluticasone propionate was prescribed as an asthma controller twice more than budesonide in this study was due to the availability of ICS molecules at the clinic and the preference of attending doctors. Montelukast was used as an add-on controller for patients with asthma on steps 3–5 or as a stand-alone controller for asthma patients with asthma on steps 1–2. In this study, the dosage of ICS/LABA was stepped down over time as recommended by the GINA strategy [7]. The proportion of patients using high-dose ICS decreased from 79.2% at the baseline visit to 27.2% after 5 years. Asthma controllers containing ICS have been shown to benefit all individuals with asthma regardless of symptom frequency [19]. ICS reduced inflammation which improves asthma symptoms control and improve lung function, its effect continues with regular use, even at lower dosage over time as in this study. A series of systematic reviews and meta-analyses showed that at low-dose ICS, patients with asthma can achieve 80–90% of the maximum obtainable benefits [20].

The proportion of patients who attended follow-up visits fell significantly after one year. This finding is in line with a finding from a national pharmacy database study in the United States [21]. Of the 5504 patients receiving an initial fill for fluticasone propionate/salmeterol combination, only 8.8% of patients had continued to refill their prescription after one year [21]. In our study, there may be several reasons for the high rate of drop-out after one year. First, many patients after receiving the correct diagnosis of asthma moved back to lower-level hospitals to receive asthma controllers which were co-paid by health insurance. Second, patients with asthma had their symptoms resolved after receiving asthma controllers and they just stopped visiting the clinic. This is the natural behavior for most patients with asthma. A systematic literature review showed that the mean level of adherence to ICS was found to be between 22 and 63% [22]. Third, out-of-pocket costs and high cost of asthma controllers may contribute to poor adherence to asthma treatment [23]. In this study, most patients were prescribed ICS/LABA instead of ICS alone as a controller at the baseline visit which is a similar trend as in Australia [24]. The reason was most patients in this study were moderate or severe asthma at the baseline visit. To increase levels of adherence to asthma controllers, doctors should make a shared decision with patients to work out the right asthma regimen for them [25,26].

There are several strong points in this real-world study. Spirometry performed at the baseline visit and at follow-up visits helped us to diagnose asthma more accurately and to evaluate treatment response more objectively [11]. One of the reasons for over- or under-diagnosis of asthma is the lack of lung function testing [27]. In the primary care unit in clinical practice, only half of all asthma diagnoses are confirmed by spirometry, and only a quarter of asthma patients are monitored annually with spirometry [28]. This study showed that the improvement of lung function was sustained over 5 years which has been much more difficult to prove in previous studies [29]. The proportion of patients with persistent airflow limitation significantly decreased from 26.7% at baseline to 12.2% at year 5 and the proportion of patients with FEV1 < 60% significantly decreased from 18.5% at baseline to 6.5% at year 5. Low pulmonary function may be due to untreated airway inflammation [30]. FEV_1_ < 60% is associated with an increased risk of exacerbations [31] and rapid decline in lung function tests [32], independent of symptom levels. The 5-year follow-up of this study demonstrated that the majority of patients with asthma do not attend the same outpatient respiratory clinic over time or the adherence to asthma controllers commonly falls over time [33]. This finding corroborates the reality gap between doctors’ expectations and patients’ behavior [34]. The five-year follow-up study showed that more work is needed to increase the proportion of patients with well-controlled asthma as the goals of GINA [1]. Among patients who adhered to asthma controllers, only 59.5% achieved well-controlled asthma after 5 years of treatment. Patients with asthma may need individualized management in which treatment decisions are driven by an objective assessment of key treatable mechanistic traits [35].

This study has some limitations. First, there were missing data because this is a retrospective and data-based extraction study. However, with the within-subject analysis of outcomes over the follow-up time, the results are still valid. Second, we do not know what happened to those without follow-up visits. However, there was no difference in baseline characteristics except for allergic rhinitis between patients who attended and patients who did not attend the follow-up visit at year 1. This finding suggests that if patients with asthma continue receiving appropriate asthma treatment, their level of asthma control will be sustained over time, regardless of their clinical characteristics.

## 5. Conclusions

In conclusion, closer detection and individualized follow-up of patients with asthma by providing them with an appropriate treatment according to GINA recommendations improved asthma symptoms controls and lung function after 3 months and the improvement was sustained over 5 years. This improvement should favorably impact asthma outcomes and alleviate its overall current burden worldwide.

## Figures and Tables

**Figure 1 jpm-13-00809-f001:**
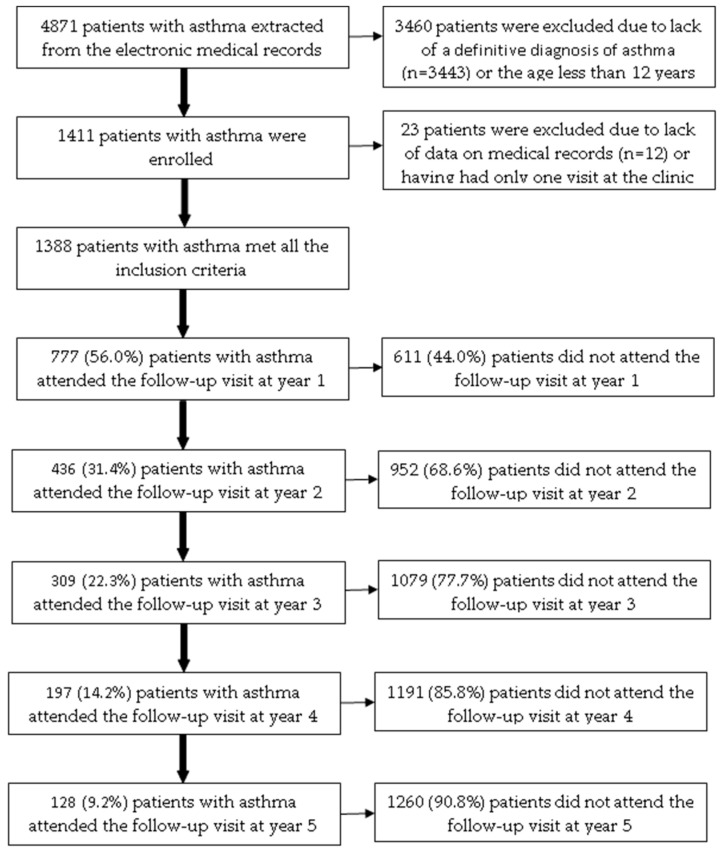
The flow chart of included patients with asthma over a five-year follow-up.

**Figure 2 jpm-13-00809-f002:**
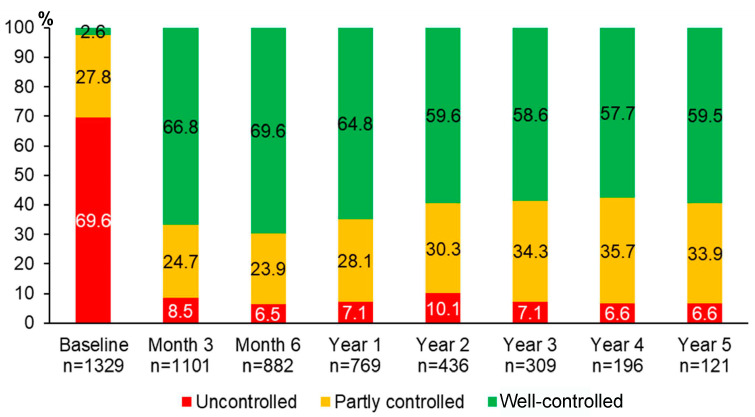
Proportions of different levels of asthma symptoms control during 5 years of asthma management.

**Figure 3 jpm-13-00809-f003:**
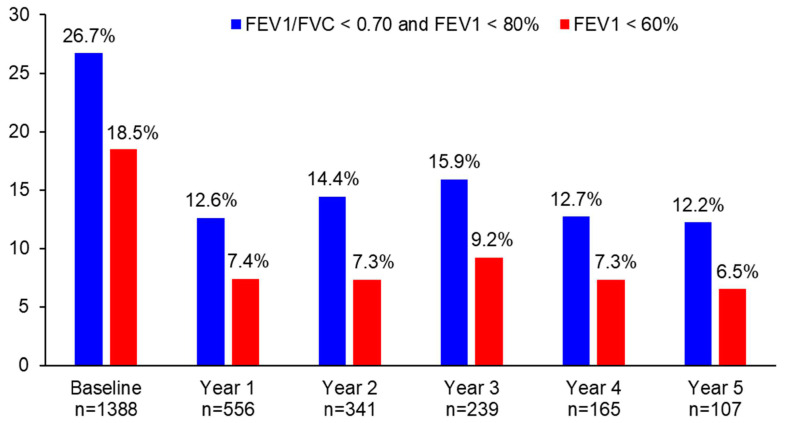
The proportion of patients with persistent airflow limitation or FEV_1_ < 60% during 5 years of asthma management.

**Table 1 jpm-13-00809-t001:** Characteristics of patients with asthma who met inclusion criteria at the baseline visit.

Baseline Characteristics	
Age (years): mean ± SD (*n* = 1388)	40.0 ± 16.7
Female (*n* = 1388)	897 (64.6%)
Education level (*n* = 106)	
High school degree or higher	72 (67.9%)
Smoking status (*n* = 1386)	
Smokers	160 (11.5%)
Non-smokers	1226 (88.5%)
Comorbidities	
Allergic rhinitis (*n* = 1388)	357 (25.7%)
BMI ≥ 25 kg/m^2^ (*n* = 1388)	206 (14.8%)
Gastroesophageal reflux (*n* = 1386)	178 (12.8%)
Childhood asthma (n = 1387)	408 (29.4%)
BMI (kg/m^2^) (*n* = 1386)	21.7 ± 3.5
Spirometry results (*n* = 1388)	
FVC (% predicted)	84.6 ± 16.9
FEV1 (% predicted)	77.5 ± 20.8
PEF (% predicted)	67.8 ± 22.4
Positive bronchodilator reversibility test (*n* = 1377) *	773 (56.1%)
Asthma severity (*n* = 1354) ^†^	
Intermittent asthma	7 (0.5%)
Mild persistent asthma	85 (6.3%)
Moderate persistent asthma	281 (20.8%)
Severe persistent asthma	981 (72.4%)
Level of asthma symptoms control (*n* = 1329) ^§^	
Well-controlled	35 (2.6%)
Partly controlled	369 (27.8%)
Uncontrolled	925 (69.6%)
Asthma controllers prescribed (*n* = 1388)	
ICS/LABA	1343 (96.8%)
Montelukast	849 (61.2%)
LAMA	7 (0.5%)
ICS molecules (*n* = 1360)	
Fluticasone propionate	919 (67.6%)
Budesonide	441 (32.4%)
Co-paid by health insurance (*n* = 1353)	324 (23.9%)

Data are presented as n (%) or mean ± SD. BMI, body mass index; FEV_1_, forced expiratory volume in one second; FVC, forced vital capacity; ICS, inhaled corticosteroid; LABA, long-acting β_2_ agonist; LAMA, long-acting muscarinic antagonist; PEF, peak expiratory flow. * Positive bronchodilator reversibility test was defined following ATS/ERS 2005 guidelines; ^†^ Asthma severity was defined by the criteria of GINA 2006; ^§^ Level of asthma symptoms control was defined by the criteria of GINA 2016.

**Table 2 jpm-13-00809-t002:** The annual rate of hospitalizations for asthma or asthma exacerbations over 5 years.

Visits	Baseline (n = 1377)	Year 1 (n = 777)	Year 2 (n = 433)	Year 3 (n = 308)	Year 4 (n = 197)	Year 5 (n = 128)
Hospitalizations	0.056 ± 0.301	0.003 ± 0.051	0.002 ± 0.048	0.0 ± 0.0	0.005 ± 0.071	0.016 ± 0.124
*p* value *		<0.0001	0.0001	0.0005	0.0049	0.1484
Exacerbations	0.120 ± 0.544	0.069 ± 0.264	0.060 ± 0.238	0.081 ± 0.285	0.056 ± 0.230	0.055 ± 0.228
*p* value *		0.0885	0.0283	0.6576	0.0309	0.0340

Data are presented as mean ± standard deviation. * *p* values for comparisons between the previous year of the baseline visit and each of the following years by paired Wilcoxon signed rank test.

**Table 3 jpm-13-00809-t003:** The improvement of lung function over 5 years of asthma management.

Spirometry Parameters	Baseline n = 1388	Year 1 n = 556	Year 2 n = 341	Year 3 n = 239	Year 4 n = 165	Year 5 n = 107
FVC, % predicted *	84.6 ± 16.9	91.1 ± 14.3 (*p* < 0.0001)	91.1 ± 15.2 (*p* < 0.0001)	91.0 ± 14.8 (*p* < 0.0001)	90.7 ± 14.5 (*p* < 0.0001)	90.1 ± 14.5 (*p* < 0.0001)
FEV_1_, % predicted *	77.5 ± 20.8	86.3 ± 17.4 (*p* < 0.0001)	86.2 ± 17.4 (*p* < 0.0001)	85.6 ± 18.2 (*p* < 0.0001)	85.9 ± 16.9 (*p* < 0.0001)	85.7 ± 16.7 (*p* < 0.0001)
FEV_1_/FVC *	75.7 ± 12.6	78.5 ± 10.3 (*p* < 0.0001)	77.8 ± 10.3 (*p* = 0.0080)	77.5 ± 10.8 (*p* < 0.0001)	77.8 ± 11.0 (*p* = 0.0384)	77.5 ± 10.2 (*p* = 0.1714)
PEF, % predicted *	67.8 ± 22.4	82.1 ± 19.7 (*p* < 0.0001)	81.5 ± 19 (*p* < 0.0001)	81.0 ± 19.5 (*p* < 0.0001)	81.3 ± 18.0 (*p* < 0.0001)	81.6 ± 16.5 (*p* < 0.0001)

Data are presented as mean ± standard deviation. * *p* values for comparisons between each of the follow-up visits and the baseline visit by paired Student t test. FEV_1_: forced expiratory volume in one second; FVC: forced vital capacity; PEF: peak expiratory flow.

**Table 4 jpm-13-00809-t004:** Relation between baseline characteristics and proportion of patients who attended the follow-up visit at year 1.

Baseline Characteristics	Follow-Up Visit at Year 1		Adjusted OR (95% CI)	*p*
Other provinces (n = 932) vs. Ho Chi Minh City (n = 456)	537 (57.6%)	232 (50.9%)	1.26 (1.00–1.60)	0.0527
Allergic rhinitis (n = 357) vs. no allergic rhinitis (n = 1031)	220 (61.6%)	549 (53.3%)	1.40 (1.09–1.81)	0.0092
BMI < 25 kg/m^2^ (n = 1182) vs. BMI ≥ 25 kg/m^2^ (n = 206)	670 (56.7%)	99 (48.1%)	1.20 (0.88–1.64)	0.2479
Co-pay (n = 324) vs. non-co-pay (n = 1029)	197 (60.8%)	561 (54.5%)	1.25 (0.96–1.62)	0.0957
Fluticasone (n = 919) vs. Budesonide (n = 441)	534 (58.1%)	224 (50.8%)	1.21 (0.95–1.54)	0.1244
High-dose ICS (n = 1050) vs. non-high-dose ICS (n = 275)	606 (57.7%)	141 (51.3%)	1.18 (0.89–1.56)	0.2497

Data are presented as n (%). BMI, body mass index; ICS, inhaled corticosteroid.

## Data Availability

All data not published within this article will be made available by request from any qualified investigator.

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
