# Peer review of "GINA Implementation Improves Asthma Symptoms Control and Lung Function: A Five-Year Real-World Follow-Up Study"

_jpm, 2023, doi:10.3390/jpm13050809_

Round 1
Reviewer 1 Report
This study, showing that an appropriate treatment according to Gina recommendations may improve asthma control, is of some interest. However it presents important limitations, as reported below.
Patients and methods
A clear description of the study design is lacking.
In particular it is unclear what the sentence (line 70) “Patients with asthma had been managed at the discretion of attending doctors” means.
In addition, it is reported (lines 96-98) that during follow-up “patients might have several visits which might not take place at the predefined time” but I did not find any description of predefined time. The timing of the visits is reported only in the results. A clear description of the timing of the follow-up visits should be included in the methods
Moreover, considering that patients underwent the 1st visit between October 2006 and October 2011, in that visit they could have been examined only according to the criteria of Gina 2006. It is unclear: 1)1st visit is the baseline visit? 2) if so why lines 132-133 report that “level of asthma symptoms control was defined by criteria of Gina 2016? 3) which criteria have been used in the follow-up visits; 4) which are the main difference between Gina 2006 and Gina 2016 criteria?
Finally, it is unclear why presence of FEV1/FVC>70% and FEV1<80% that usually define “airflow limitation” has been used to define “persistent airflow limitation”. This should be clarified
Results
Figure 1 show that the number of patients attending each visit markedly decreases over time, from 1329 at baseline to 121 at year 5. Although a loss of patients is common in this type of study, a 10 fold reduction is quite surprising and more importantly, could have influenced the results. It could be of interest to show (at least in the supplementary materials) the results of each of the 8 visits of the 121 patients who completed the follow-up.
A similar comment is valid for results in figure 2 and tables 2 and 3.
Finally I would like to report that I was not able to see the supplementary materials (Figure S1 and Table S1)
Reviewer 2 Report
Minor Comments
In this research article, the authors investigate the level of asthma symptoms control and lung function over 5 years of GINA (Global INitiative for Asthma) implementation. The manuscript is well-written, and the study design and data analysis are robust.
Point 1: The experimental method part is clear; it would be better to draw a detailed workflow, added as figure 1. Otherwise, it would be difficult for the reader to capture the overall picture of the study.
Point 2 : In sentence 271; space is missing between “symptomscontrols”. Rewrite sentences 195-196 “These results…..persisted for one year”.
Point 3: A few minor grammar mistakes need to be corrected thoroughly.
Point 4: Overall, I could not fault the experiments or the interpretation. However, future experimental investigations would be informative.
Minor editing of the English language required
